# Multi-context modeling of driver pathways reveals common and specific mechanisms across 23 cancer types

**Wenjia Zhou**[1,2], **Junhua Zhang** [1]*

**1** CEMS, NCMIS, RCSDS, Academy of Mathematics and Systems Science, Chinese Academy of Sciences, Beijing, China, **2** School of Mathematical Sciences, University of Chinese Academy of Sciences, Beijing, China

* zjh@amt.ac.cn

## Abstract

Discovery of cancer driver pathways is essential for targeted therapies, since these pathways govern tumor progression and treatment resistance. However, their context-specific patterns across populations remain poorly understood. Leveraging pan-cancer genomic data, we apply our two models, EntCDP and ModSDP, to perform stratified analyses from four perspectives: region, tumor type, age group, and risk factors. Our results reveal the regional biases in perturbed pathways, such as PI3K-Akt in Chinese patients and GPCR in American patients with bladder cancer. Subtype comparisons highlight the mTOR signaling in lung adenocarcinoma and the FoxO signaling in lung squamous cell carcinoma. Pediatric-adult comparisons emphasize the enrichment of Ras signaling in pediatric acute myeloid leukemia and PAK signaling in pediatric glioblastoma, respectively. Risk factor associations further link Notch-mediated pathways to alcohol consumption and CDKN-regulated pathways to obesity-related cancers. Our findings demonstrate the utility of stratified driver pathway analysis in uncovering common and specific mechanisms, which can help prioritize context-aware therapeutic targets.

## Author summary

Cancer develops as a result of disruptions in signaling pathways driven by gene mutations. However, the heterogeneity of these pathways across patient groups with distinct clinical characteristics remains inadequately understood. Here, we aim to bridge the existing knowledge gap by utilizing our newly developed models: EntCDP and ModSDP. These models are applied to analyze gene mutation data and clinical information from 55 cohorts, encompassing 23 different cancer types across various platforms. We explore how geographic factors drive regional biases, uncover differences among cancers with similar functions or locations (e.g., digestive system cancers, lung subtypes, sex-specific cancers, and metastatic tumors with preferred sites), and examine age-related shifts

**Data availability statement:** Somatic mutation dataset in this study can be downloaded from the open platform including TCGA

(https://portal.gdc.cancer.gov/), ICGC (https://dcc.icgc.org/releases/current/Projects/), PCAWG (https://xenabrowser.net/datapages/?hub=https://pcawg.xenahubs.net:443) and cBioPortal (https://www.cbioportal.org/). The code for the Matlab package of EntCDP and ModSDP is freely available at https://github.com/zjh136/Project–EntCDP-ModSDP.

**Funding:** This work was supported by the National Key Research and Development Program of China (2022YFA1004800 to JZ). The funders had no role in study design, data collection and analysis, decision to publish, or preparation of the manuscript.

**Competing interests:** The authors have declared that no competing interests exist.

to reveal developmental and physiological impacts. Additionally, we investigate how lifestyle factors, such as smoking, alcohol consumption, and obesity, influence signaling pathways through environmental exposures. These findings offer a comprehensive framework for understanding the heterogeneity of cancer signaling across diverse populations and clinical contexts. Our study not only elucidates the mechanisms underlying cancer progression, but also identifies potential targets for group-based therapeutic strategies.

## Introduction

Dysregulation of pathways associated with human cancers can lead to abnormal cell proliferation and tumorigenesis. One underlying mechanism driving this process is genetic mutations in key components of pathways [1,2]. High-throughput sequencing technologies have revealed abundant mutational landscapes across various types of cancer [3–5], with heterogeneity widely observed [6,7]. This heterogeneity is manifested by a larger number of genes that are infrequently altered in certain types of cancer, rather than by a small number of universal genes that are frequently altered across multiple tumors [8]. Therefore, a persistent challenge lies in identifying the driver genes that promote tumorigenesis from a multitude of mutant genes across different tissues, and further elucidating how these genes function at the level of signaling pathways.

Existing approaches for identifying driver pathways or modules can be classified based on interaction knowledge, mutation frequency and pathway characteristics, often utilizing optimization models or network-based methods [9,10]. In terms of the patterns through which genes constitute a signaling pathway, the concepts of coverage and mutual exclusivity were described as two key features. The underlying evidence is that tumors gain a sufficient selective advantage as long as a single alteration disrupts the same functional pathway. In this context, Vandin et al. [11] developed the Dendrix (de novo driver exclusivity) method to maximize a weight function characterized by coverage and exclusivity of pathways. Their Markov chain Monte Carlo (MCMC) approach was later improved by an efficient binary linear programming (BLP) model [12]. Furthermore, the ComMDP and SpeMDP [13] models were developed at the pan-cancer level to discover common gene sets shared across distinct tumor types and specific gene sets of certain one or multiple tumor types compared to other types, respectively. To address the limitations of ComMDP and SpeMDP in identifying cancer-associated signaling pathways, we recently developed two refined models: EntCDP (Entropy-based Common Driver Pathway) and ModSDP (Modified Specific Driver Pathway) [14]. EntCDP leverages information entropy to balance the trade-off between coverage and exclusivity in ComMDP. ModSDP improves upon SpeMDP by better accounting for exclusivity. These comparative analyses naturally pave the way for cancer research and personalized medicine: commonality can guide the adoption of a consistent drug regimen across different cancers, while specificity enables the development of targeted therapies for individual cancer types. Therefore, it is necessary to conduct pan-cancer research from different perspectives.

On the other hand, genes serve as carriers of signaling pathways and can lead to cancer through mutations that disrupt these pathways, but not all genes are associated with cancer. To identify oncogenic mutations capable of driving malignancy, Martínez et al. proposed the Integrative OncoGenomics (IntOGen) framework [15] and compiled a comprehensive compendium of 568 driver genes across most tumor types. However, there is a lack of understanding regarding how these genes function as components of signaling pathways. This inspired us

to narrow our search for signaling pathways to these driver genes, which are more biologically significant.

Many pan-cancer projects have provided extensive data on gene mutations and advanced the field of pan-cancer genome research [16–18]. However, few studies have utilized clinical information to compare the signaling pathways across multiple cancer types under different conditions. To address this gap, we collected gene mutation data as well as sample attributes of 23 types of cancer from multiple platforms (TCGA, ICGC, MSKCC, TARGET, etc.), and investigated common and specific pathways using EntCDP and ModSDP in patients classified based on the following four conditions: (1) For patients diagnosed with the same cancer but originating from different nations or regions (such as CN, AU, US, etc., 12 in total), we identified novel targeted drivers for treating the relevant cancers based on common findings, and highlighted therapeutic significance of pathway prediction derived from region-specific findings; (2) For cancers sharing similar functionality or anatomical location, we analyzed common and specific signaling characteristics among comparative cancers, including cancers of the digestive system, lung subtypes, sex-specific cancers, and cancers prone to metastasis along with their metastatic sites; (3) For children who developed malignant disease, we sought to identify biomarkers for subtypes of hematological and neural tumors. We also aimed to uncover the specific characteristics of glioblastoma multiforme and acute myeloid leukemia through comparisons with adults; (4) For patients with or without risk factors such as smoking, drinking and obesity, we aimed to determine the signaling pathways disrupted by these factors that trigger cancer. Furthermore, we sought to identify additional driver factors beyond the known risk factors. In summary, our study provides novel insights into targeted cancer therapies in different regions, ages, and risk factors, thereby advancing the field of personalized cancer treatment.

## Results

### Sample characteristics

To facilitate a comparative analysis of signaling pathways among multiple cancer types under different conditions, we collected tumor genomic mutation profiles from 55 cohorts representing 23 typical types of cancer. These cohorts comprised 14,564 adult and 1,539 pediatric tumor samples from large-scale cancer genomics programs, including The Cancer Genome Atlas (TCGA) [2], the International Cancer Genome Consortium (ICGC) [19] and cBioPortal [9] (Fig 1A and S1 Table). After the exclusion of silent mutations, we selected samples with at least one alteration event (e.g., somatic mutations, small insertions/deletions, copy number variations, etc.) for further analysis.

In addition, we collected clinical data for these samples to investigate clinicopathological characteristics, including region, age, smoking history, drinking history, and obesity. However, the paired clinicopathological data were not always complete and comprehensive (see **Materials and methods**). Among the 55 cohorts, 50 had nationality or regional attributes, while the remaining 5 cohorts were primarily sourced from the cBioPortal. For glioblastoma multiforme (GBM) and acute myeloid leukemia (AML) cohorts, all samples in cohorts LAML, GBM and GBM_MSKCC were from individuals aged 18 years or older, except for LAML_CN, which did not provide age information. To ensure that only adult samples were included in the LAML_KR cohort for comparison with the pediatric cohort, we excluded 3 patients under the age of 18 from the total of 205 samples. TCGA provided the most comprehensive exposure data for risk factor analysis: 96.48% (1426/1478) of the LUAD, HNSC and BLCA samples were used to examine the role of smoking as a risk factor; 96.46% (817/847) of the ESCA, HNSC and PAAD samples were used to assess the effects of drinking; and 66.18%

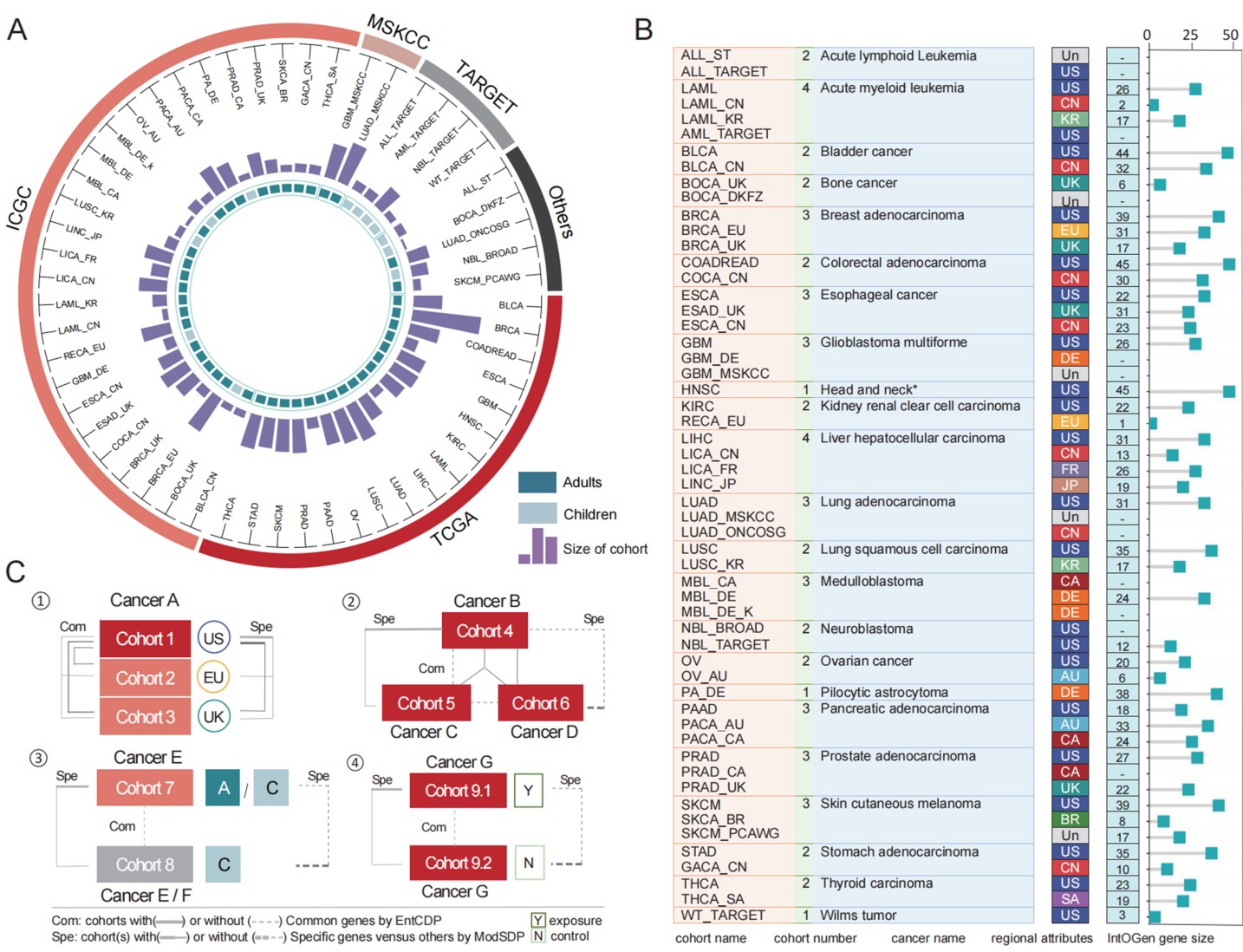

**Fig 1. Datasets of genetic mutations and framework of analysis.** (A) Summary of tumor genetic mutation datasets collected from public platforms. The outer ring indicates the sources of all cohorts included in this study, which are connected to their respective names through T-shaped crosses. The histogram inside illustrates the sample size of each cohort, and the innermost ring indicates whether the cohort mainly consists of adults or children. (B) Each column of the table provides detailed information about all 55 cohorts, including cohort names, the number of cohorts for each cancer type, cancer types, regional attributes, and the number of IntOGen genes for each cohort. The lollipop diagram on the right illustrates the numbers in the last column of the table. Different colors are used to distinguish different countries or areas. Un: unknown. -: no records. (C) Four parts of our analysis. Wires connecting different cohorts represent comparisons conducted using EntCDP (common pathways, Com) and ModSDP (specific pathways, Spe). The background colors of cohorts correspond to the platforms shown in (A). Head and neck*: Head and neck squamous cell carcinoma. A: adults; C: children; Y: exposure groups; N: control groups.

(595/899) of the COADREAD and LIHC samples were included in the analysis of obesity-related risks.

We focused our search on a well-curated collection of 568 mutational cancer genes identified by IntOGen [15], which were derived from somatic mutations data of approximately 28,000 tumors across 66 cancer types. As nearly 80% (42 out of 55) of our cohorts can overlap those used in IntOGen (Fig 1B), the cancer genes identified by IntOGen within these cohorts serve as an important benchmark for validating our results. Therefore, we used EntCDP and ModSDP to identify and compare the driver pathways across 23 cancer types based on the

568 genes, providing insights into the associations between these pathways and region, age and risk factors (Fig 1C).

## EntCDP and ModSDP reveal regional bias in driver genes of the same cancer

**Robust identification of driver genes via pathway models.** Before the investigation of pathway findings, we proposed a brief test to verify the reliability of the results derived from EntCDP and ModSDP models. The driver genes discovered by IntOGen provide insights into the mutational features of each cohort (Fig 1B and S1 Table) and provide a valuable benchmark for comparison with our findings. The main motivation is to examine, using the hypergeometric test, the degree of overlap between common (or specific) IntOGen genes of comparable cohorts and their common (or specific) driver gene sets discovered by EntCDP (or ModSDP) (Sect A in S1 Text). Since some cohorts lack IntOGen genes, we restricted the hypergeometric test to cohorts containing IntOGen genes (Sect B in S1 Text).

In spite of the incompleteness of IntOGen driver genes, those reported in the literature and associated with cohort characteristics, such as cancer type and geographic location, can serve as a credible reference for validating novel discoveries. In a sense, genes not assigned to any cohort by IntOGen and lacking annotation in the literature can be considered new candidate driver genes for a certain cancer.

**Targeting novel driver genes in region-specific signaling pathways.** The information on nationality or sequencing location in the collected data enables us to geographically stratify samples with the same cancer type (Fig 1B). Here, we focus on cohorts of each cancer type and apply EntCDP and ModSDP to cohorts of the same type but different geographic origins.

There is some overlap between the genes discovered by our models and those cohort-sensitive genes (i.e., IntOGen driver genes). At least 20% of common genes identified by EntCDP in each cohort group (Fig 2A) are IntOGen driver genes, all except kidney cohorts KIRC and RECA_EU, which share only one IntOGen gene (S2 Table). In addtion, significant overlap between IntOGen and ModSDP can be seen in half (7/14) types of cancer (S3 Table).

While IntOGen overlapping driver genes can confirm the accuracy of our results, non-IntOGen driver genes can either be validated as drivers or predicted to serve in such a capacity. For example, two common genes, *TP53* and *PIK3CA*, identified in breast adenocarcinoma cohorts from the US, EU and UK, are IntOGen genes shared by all three cohorts (colored by dark purple in Fig 2A), while *GATA3* is a known IntOGen gene present in two of the cohorts (colored by light purple). The three genes function in the IGF1 receptor (IGF-1R) signaling pathway and are identified by EntCDP when the size of the search set (parameter $K$) is 3 (Fig 2C and Table S1.3 in S1 Data). It has been reported that the IGF-1R pathway plays an important role in gynecologic cancers such as breast tumors [20]. When $K = 10$, we identified other seven genes mutually exclusive with the three genes. Although *H3F3A*, *CARS*, *GTF2I* and *SEPT9* are not IntOGen genes of the three cohorts (colored by yellow), they have been proved to play important roles in the development of breast adenocarcinoma [21–24]. For example, *SEPT9*, a septin gene encoding multiple isoform, has been confirmed as a promising tool in breast cancer detection [24]. However, those not previously reported in the literature (colored by pink), including *TAF15*, *HIST1H4I* and *FAM46C*, may facilitate the identification of novel targets for the treatment of breast adenocarcinoma. Moreover, there is a series of driver genes specific to the cohort BRCA relative to BRCA_UK and BRCA_EU, with gene set {*CDH1*, *MAP2K4*, *PIK3R1*, *MAP3K1*} detected by ModSDP when

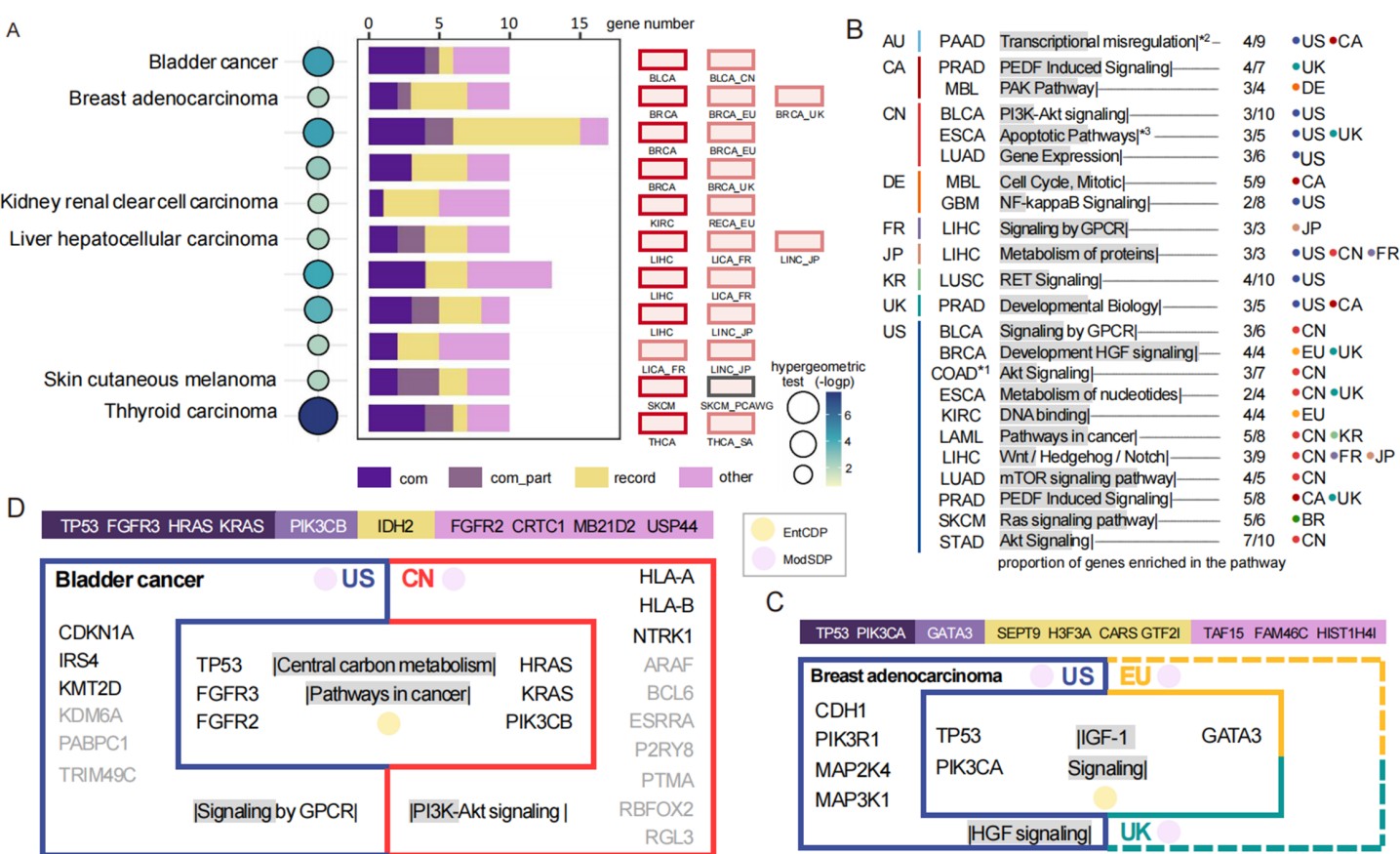

**Fig 2. Results of the reliability test and regional differences in signaling pathways.** (A) The hypergeometric test reveals significant results for six types of cancer with IntOGen genes in each cohort, which investigates the overlap between cohorts' common genes identified by IntOGen and EntCDP. The size of the circles corresponds to the significance level. Different colors in the bar graph present different features of genes identified by EntCDP: dark (light) purple indicates common genes obtained from all (part of) selected cohorts by IntOGen; yellow (pink) denotes non-IntOGen genes of any selected cohort that have (not) been verified as drivers in the literature. (B) Specific pathways of patients with a certain cancer type coming from one country (left) relative to other counties (right, denoted by colorful circles). The gray-filled bar for each signaling pathway indicates the proportion of identified genes enriched in that pathway, as shown by the ratio on the right. (C, D) Common signaling pathways among, and pathways specific to, patients with breast adenocarcinoma from the United States versus those from European Union countries and Britain (C), or with bladder cancer from the United States versus China (D). The dashed line indicates no significant results. The bar above clearly lists the names of genes with different characteristics from (A). *1: COADREAD, colorectal cancer; *2: Transcriptional misregulation in cancer; *3: Apoptotic pathways in synovial fibroblasts.

$K = 4$ (Table S1.6 in S1 Data). The four genes are members of the development HGF signaling pathway (Fig 2B and 2C), which can provide valuable diagnostic, prognostic and predictive biomarkers for personalized cancer therapy [25]. Targeting this pathway may provide clinical benefits for breast cancer patients from the United States.

Significant overlap between common genes discovered by both IntOGen and EntCDP is also observed in bladder tumors ($p = 0.0001$). *FGFR3*, *HRAS* and *KRAS*, along with known oncogene *TP53*, are typical biomarkers for bladder cancer associated with relevant pathways, and are discovered by EntCDP when $K = 4$ (Table S1.1 in S1 Data). When $K = 10$, EntCDP detected another six genes mutually exclusive with these four genes. Among them, *PIK3CB* is an IntOGen gene of the bladder cohort from America but not China, and *IDH2* is not present in any cohort but has been reported to be of potential therapeutic significance [26]. The remaining four genes, *CRTC1*, *MB21D2*, *USP44* and *FGFR2*, may drive bladder cancer through other mechanisms of pathway crosstalk, which requires further investigation.

A gene set comprising 10 genes is the only significant gene set specific to BLCA_CN compared to BLCA, among which *NTRK1*, *HLA-A* and *HLA-B* are associated with the PI3K-Akt signaling pathway (Fig 2C and 2D and Table S1.2 in S1 Data). In contrast, American patients with bladder cancer tend to harbor mutations that disrupt signaling by GPCR, with *CDKN1A*, *IRS4* and *KMT2D* identified in the 6-gene set. Apart from these genes in specific pathways, some light gray genes in Fig 2D, such as *ARAF*, *ESRRA* and *PTMA* (from the China cohort), as well as *KDM6A*, *PABPC1* and *TRIM49C* (from the American cohort), are predicted by ModSDP as region-specific drivers of bladder cancer and require further experimental validation.

Common pathways shared by patients from different countries can provide novel targeted drivers for cancer treatment. Meanwhile, the therapeutic significance of pathway prediction is embodied within specific findings distinguished among areas, as shown in Fig 2B. There are also differences in the specific signaling pathways among different cancer types in the same region, which emphasizes the need to distinguish both regional and cancer-type differences in cancer treatments.

## EntCDP and ModSDP provide pathway analysis of tissues with similar locations or related functions

Generally, different types of tumors have distinct mutation profiles and therefore require different treatments. However, certain types of cancer share commonalities and connections, particularly in terms of function and anatomical location. To decipher potential commonalities and differences among various cancers, we primarily utilized mutation data from TCGA to mitigate the influence of confounding variables. We selected several comparable groups for analysis, including (1) esophageal carcinoma (ESCA) and gastric carcinoma (STAD); (2) lung squamous cell carcinoma (LUSC) and lung adenocarcinoma (LUAD); (3) colorectal cancer (COADREAD) and liver hepatocellular carcinoma (LIHC); (4) breast adenocarcinoma (BRCA) and ovarian cancer (OV); and (5) kidney renal clear cell carcinoma (KIRC) and prostate adenocarcinoma (PRAD).

**Signal in esophageal and gastric carcinoma.** The esophagus and stomach, two crucial organs of the digestive system, are closely related. However, there is a lack of studies on tumor biomarkers through genotype similarity analysis. Therefore, understanding the etiologies of ESCA and STAD at the pathway level will facilitate screening and prevention efforts in high-risk populations.

By applying EntCDP to cohorts ESCA and STAD, we identified a common driver gene set when $K = 9$ (Fig 3A top and Table S2.6 in S1 Data). Among them, *ARID1A*, *TP53*, and *CDH1* are also IntOGen genes in both cohorts. Although not listed by IntOGen for these cohorts, *ACVR1* and *CXCR4* have been implicated in the tumorigenesis of both cancers [27,28]. The remaining genes, including *LDB1*, *PCBP1*, *CDK4*, and *UBE2D2*, are novel predictions by EntCDP and may inform future therapeutic strategies. Enrichment analysis further supports the relevance of these genes, since *TP53*, *CDH1*, *CDK4*, and *CXCR4* are typical representations of pathways in cancer (Fig 3B).

To investigate specific signals, we further applied ModSDP to identify distinct driver genes between cohorts ESCA and STAD. We identified *PRKAR1A*, *SETD1B* and *TP53* as ESCA-specific driver genes, which are essential for the metabolism pathway. In contrast, *ARID1A*, *NFE2L2* and *TP53* were recognized as STAD-specific genes and serve as major drivers in the gastric cancer pathway (Fig 3A bottom and Table S2.7 in S1 Data). Notably, *TP53* plays a prominent role in the development of esophagogastric cancers, not only because of its high mutation frequency, but also due to its enrichment in multiple cancer-related

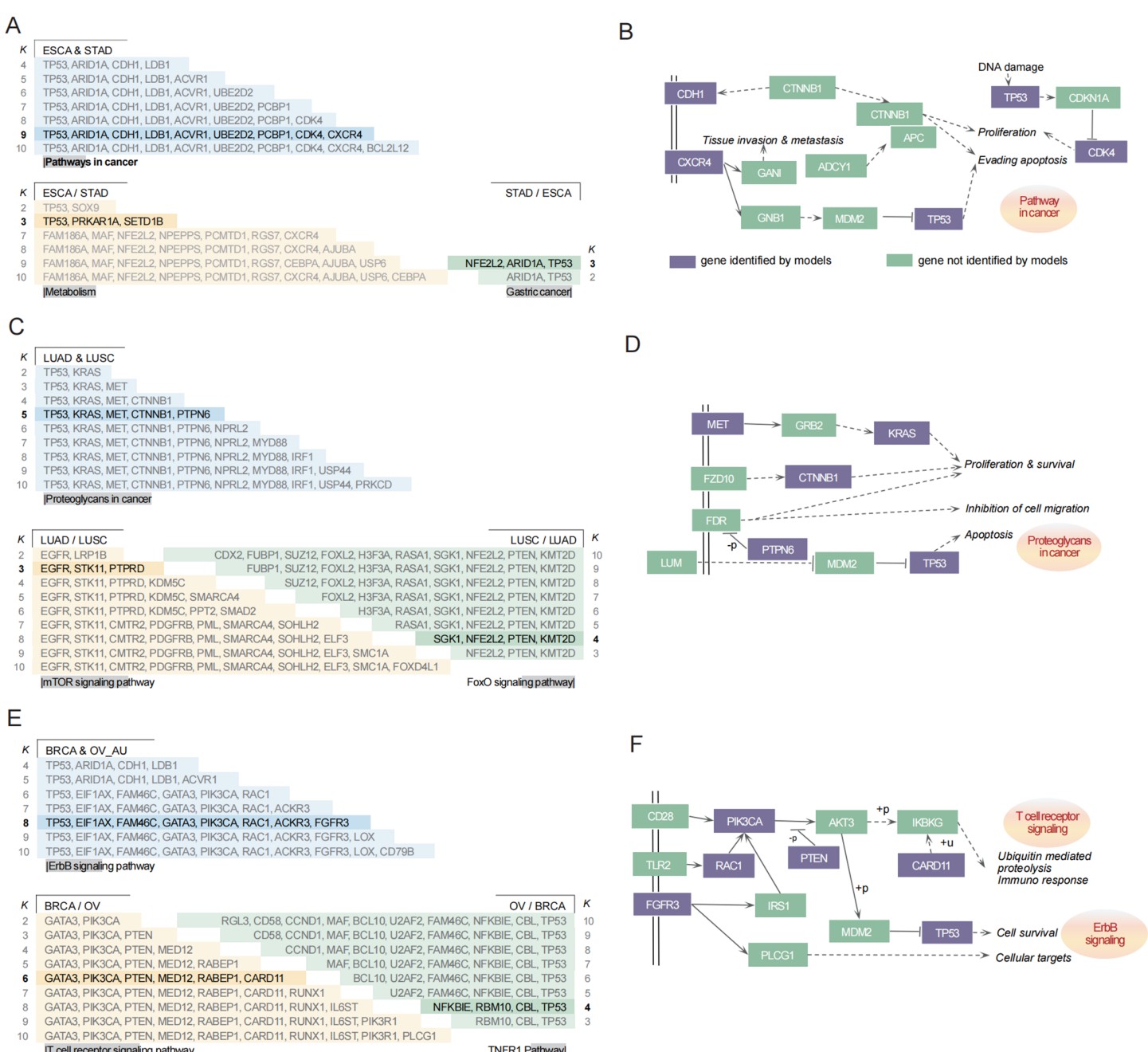

**Fig 3. Pairwise comparison of cancer types similar in location or function.** (A, C, E) The top panels show the common gene sets between two cancers identified by EntCDP (A & B), and the bottom panels display the specific gene sets of cancer A relative to cancer B (A / B) or vice versa (B / A) by ModSDP. Only significant sets are shown. Numbers in the leftmost or rightmost columns denote the parameter K, which means the number of genes in each set. Genes in the highlighted rows are enriched in the signaling pathway at the bottom of each ladder table. (B, D, F) Regulation of genes involved in the signaling pathways shown in (A, C, E), respectively. Genes marked in purple are identified by EntCDP or ModSDP, while genes in green are not.

pathways. Therefore, we hypothesize that *TP53* may be key to resolving gastrointestinal disorders.

**Pathways involved in non-small cell lung cancer.** LUSC and LUAD are two major subtypes of non-small cell lung cancer (NSCLC) but demonstrate distinct genetic drivers and

divergent prognostic outcomes [29]. Despite some success in the conventional treatment for lung cancer, subtype-specific targeted therapies remain necessary to understand genetic heterogeneity among subtypes.

The biological relevance of EntCDP is supported by the identification of the common gene set {*CTNNB1*, *KRAS*, *MET*, *PTPN6*, *TP53*} in cohorts LUAD and LUSC (Fig 3C top). Among them, *KRAS* is an IntOGen gene of cohort LUAD, whose mutation is also a critical criterion in targeted therapy for patients diagnosed with LUSC [30]. Additionally, this gene set is enriched in the proteoglycans in cancer pathway (Fig 3D). In a combined experiment [31], the expression of proteoglycan genes significantly decreased in tumors compared to normal lung tissue, aligning with our findings and suggesting their potential as promising biomarkers.

As for the specificity, we identified a LUAD-specific gene set: {*EGFR*, *STK11*, *PTPRD*}, among which *EGFR* and *STK11* are involved in the mTOR and PI3K-Akt signaling pathways (Fig 3C bottom). It has been reported that dioscin, a therapeutic agent for lung adenocarcinoma, can suppress the proliferation, invasion and EMT of lung cells via the inactivation of AKT/mTOR/GSK3$\beta$ signaling pathway [32]. Notably, *PTPRD* is a novel candidate predicted by ModSDP within LUAD-specific gene set and may represent a promising target for future experimental validation. By contrast, the LUSC-specific gene set includes {*KMT2D*, *NFE2L2*, *PTEN*, *SGK1*}. Among these, *KMT2D* and *NFE2L2* are key regulators of LUSC tumorigenesis [33,34] and well-established components of the FoxO pathway. Therefore, we predict that *PTEN* and *SGK1* may serve as effective therapeutic targets for LUSC by mediating the FoxO signaling pathway. Collectively, these findings demonstrate the ability of ModSDP to recover biologically meaningful and therapeutically relevant gene sets specific to cancer subtypes.

**Typical malignant tumors in males and females.**  Sex differences in cancer arise from the effects of circulating sex hormones [35], which influence organs unique to or significantly different between males and females.

In the previous section, we focused on breast cancer due to its high prevalence among women in many countries. Here, we compared ovarian cancer and breast carcinoma given their close association [36]. We found no common gene sets between cohorts BRCA and OV; however, we surprisingly identified significant commonalities between cohorts BRCA and OV_AU ($p$ = 0.0016). For $K$ = 8, the ErbB signaling pathway emerged as a common pathway, characterized by *TP53*, *PIK3CA*, *RAC1* and *FGFR3* (Fig 3E top and 3F). Among these, *TP53* and *PIK3CA* are IntOGen genes of both cohorts, while the roles of the other six genes, *ACKR3*, *EIF1AX*, *FGFR3*, *LOX*, *RAC1*, and *GATA3*, have also been highlighted in studies of both ovarian cancer and breast carcinoma [37–47]. These findings highlight the biological relevance and robustness of the identified gene set.

For the differences, BRCA-specific analysis highlighted the T cell receptor (TCR) signaling pathway, whereas OV was specifically characterized by the TNFR1 pathway activation (Fig 3F). In BRCA, we identified *CARD11*, *PIK3CA*, *PTEN*, and *GATA3* as key genes in the TCR pathway when $K$ = 6 (Fig 3E bottom). Gu-Trantien et al. examined the prognostic factor of CD4+Tfh cells and the function of the T cell receptor signaling pathway in human breast cancer [48]. Conversely, we detected {*TP53*, *CBL*, *NFKBIE*, *RBM10*} as OV-specific drivers, and the first three are components of the TNFR1 pathway that exerts indirect cytotoxic effects on ovarian cancer stem cells [49]. Furthermore, we identified novel predictions in these contexts, including *MED12* and *RABEP1* in BRCA, and *MED12* as an OV-specific candidate for further functional validation.

It seems that pathway analysis for the two male urological cancers is less specific, as the specific pathways identified are the pathways in cancer for KIRC and the transcriptional misregulation in cancer for PRAD, respectively. See Sect C in S1 Text for more details.

Beyond the four cancer type pairs detailed in our study, other comparable groups may also be informative in future clinical trials (Tables S2.11–S2.17 in S1 Data).

## EntCDP and ModSDP display the signaling profile for children

In recent years, reports on pediatric tumors have increased rapidly and attracted widespread public attention. Although advanced technology has profoundly reduced the mortality rates of pediatric cancer, the threat of death and ineffective treatment continues to cast shadows over too many vulnerable children [50].

Focusing on prevalent childhood tumors, we analyzed both commonality and distinctions among different pediatric tumors and their distinctions from adult tumors. We first presented a comprehensive comparison between glioblastoma multiforme (GBM) and acute myeloid leukemia (AML), both of which pose threats to children and adults, from both cross-sectional and longitudinal perspectives (Fig 4A). Subsequently, we discussed the biomarkers for two subtypes of hematological tumors and four subtypes of neural tumors.

**Comparison of adult and pediatric cohorts in AML and GBM.**  The prominence of GBM among brain tumors underscored by its high morbidity and mortality rates in both children and adults [50]. Meanwhile, AML, a representative liquid tumor, is the most frequently diagnosed leukemia in adults (25%) and accounts for 15–20% of cases in children [51].

By leveraging somatic mutation data from both adult and pediatric cohorts of GBM and AML, EntCDP successfully identifies biologically meaningful common gene sets between two age groups. Specifically, for GBM (adults) and GBM_DE (children), four genes (*CASP8*, *PIK3CA*, *PTEN*, *LPAR4*) related to pathways in cancer are identified (Fig 4A right A & C and Table S3.1 in S1 Data). For LAML (adults) and AML_TARGET (children), the common pathway is cytokine signaling in the immune system, comprising six genes: *TP53*, *FLT3*, *KIT*, *KRAS*, *NRAS*, and *PTPN11* (Fig 4A left A & C and Table S3.6 in S1 Data). This finding is consistent with the fact that abnormalities in cytokine signaling pathways are characteristic of all types of leukemia [52].

Age-specific differences in signaling pathways can guide the discovery of therapeutic targets tailored to pediatric and adult cancers. In GBM, *CEBPA* and *ID3* in the NF-$\kappa$B signaling pathway (Fig 4A right C/A) are discovered as specific genes of children compared to adults when $K = 8$ (Table S3.4 in S1 Data), which may serve as potential targets in childhood glioblastoma. The adult-specific signaling is the MAPK signaling pathway, regulated by *TP53*, *FGFR4*, *EGFR* and *IRAK1* (Fig 4A right A/C), which has been reported to promote glioblastoma malignancy [53]. Treatment for AML children may rely on downstream effectors of receptor tyrosine kinase (RTK), such as the Ras and phospholipase D signaling pathways [54], since we discovered *NRAS*, *KIT*, *PTPN11* and *KRAS* in the two pathways when comparing AML_TARGET (children) to LAML (adults), LAML_CN (adults) and LAML_KR (adults) at $K = 4$ (Fig 4A left C/A and Table S3.7 in S1 Data). When $K = 8$, we detected {*ARHGAP35*, *CEBPA*, *DNMT3A*, *ASXL1*, *PHF6*, *SUZ12*, *TNC*, *MTOR*} as an adult-specific gene set for cohort LAML (Table S3.10 in S1 Data), among which *CEBPA*, *MTOR*, *SUZ12* and *ARHGAP35* are core components of the developmental biology pathway (Fig 4A left A/C). All remaining genes except for *TNC* have been previously implicated in AML [55–57]. Thus, we predict *TNC* as a novel candidate gene identified by ModSDP.

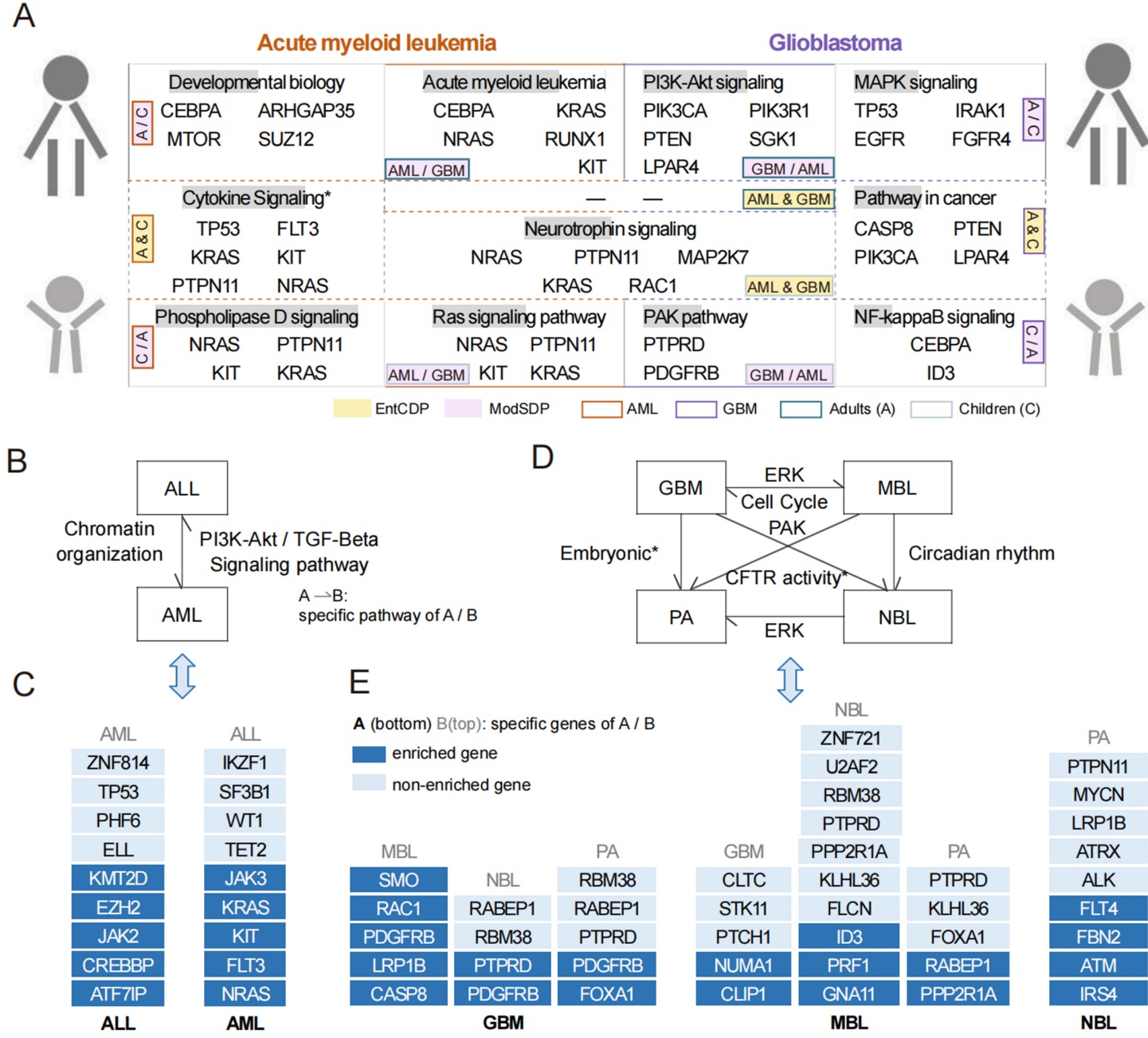

**Fig 4. Interpretation of signaling pathways in pediatric and adult tumors.** (A) Common and specific pathways of acute myeloid leukemia and glioblastoma in adults and children. (B, D) Specific signals of two hematopoietic tumors (B) and four neural tumors (D). The pathways labeled on directed edges represent significant signals specific to one cancer at the tail compared to another at the arrowhead. (C, E) Corresponding to (B) and (D), genes enriched in identified pathways are stacked from bottom to top and marked in dark blue. The remaining genes, depicted in light blue, are also identified by ModSDP but are not explicitly annotated for enrichment in the corresponding signaling pathway. Both dark blue and light blue genes are displayed under the same parameter $K$. Cytokine signaling*: Cytokine signaling in immune system; Embryonic*: Embryonic and induced pluripotent stem cells and lineage-specific markers; CFTR activity*: Regulation of CFTR activity.

**AML and GBM comparison within pediatric and adult groups.** Next, we summarized some meaningful outcomes from the comparison between AML and GBM conducted separately in children and adults.

The results show that the neurotrophin signaling pathway [58] is the common factor driving tumor development in pediatric patients from the cohorts AML_TARGET and GBM_DE_K. However, no gene sets were detected from mutations in adults by EntCDP, which indicates greater cancer heterogeneity observed in adult patients (Fig 4A middle and Table S3.11 in S1 Data).

The distinct pathways identified between adult cancers demonstrate the biological significance of ModSDP. As illustrated in the upper part of Fig 4A, we successfully identified *CEBPA*, *NRAS*, *KIT*, *KRAS* and *RUNX1* as specific genes for adults with AML compared to those with GBM, and these genes are typical members of the acute myeloid leukemia pathway. On the other hand, adults with GBM are specifically affected by the PI3K-Akt signaling pathway (Table S3.12 in S1 Data), inhibition of which has been confirmed to be effective in clinical trials [59]. For children (lower part in Fig 4A), Ras and PAK signaling pathways are specifically associated with AML and GBM, respectively (Table S3.13 in S1 Data). As a tumor suppressor gene in the PAK pathway, inactivated *PTPRD* may lead to increased cell growth and suppression of apoptosis in GBM [60]. Mutations in the *PTPN11*, which encodes a phosphatase and regulates the Ras signaling pathway, are well characterized in children with AML [61,62]. Therefore, alterations in these two signaling pathways may have important implications for potential anticancer treatments.

In summary, the PI3K-Akt signaling pathway (*PIK3CA* and *PTEN*) is critical in adults with GBM, while the PAK signaling (*PTPRD* and *PDGFRB*) is specific to children with GBM. In AML, *KIT*, *NRAS*, and *KRAS* constitute core drivers, with *CEBPA* and *PTPN11* further distinguishing adult and pediatric cases, respectively. These genes represent key molecular drivers within their respective biological contexts. Other genes, such as *TNC* in adults with AML, and *CEBPA* and *ID3* in children with GBM, are proposed in our study as predicted therapeutic molecular targets within their associated signaling pathways.

**Biomarkers for subtypes of pediatric hematological and neural tumors.**   The final section focuses on two subtypes of leukemia: acute lymphoid leukemia (ALL) and acute myeloid leukemia (AML), as well as four subtypes of neural tumors: glioblastoma (GBM), neuroblastoma (NBL), medulloblastoma (MBL) and pilocytic astrocytoma (PA).

Both ALL and AML are common malignancies in children and young adults, but they differ in biology characteristics and clinical responses to various chemotherapies [63]. When analyzing similarities between the two subtypes, we found a set of 10 common genes shared by ALL_ST and AML_TARGET (Table S3.19 in S1 Data), among which *TP53*, *NRAS*, *KIT*, *NF1*, *PTPN11* and *KRAS* are involved in the RET signaling. It is consistent with previous reports that AML cells exhibit activation of the RET signaling pathway [64]. Although there are few related reports for ALL, our results predict novel therapeutic targets, such as *WT1* and *FLT3*. Additionally, the two subtypes, ALL_TARGET and AML_TARGET, differ in signals dominated by specific genes shown in Fig 4B and 4C. Surprisingly, we noticed that *JAK2* and *JAK3*, two members of Janus kinase (JAK) family, are distinguishing markers for ALL and AML, respectively (Fig 4C).

Central nervous system tumors are the leading cause of cancer-related death in children [65]. In addition to GBM mentioned above, MBL and PA are particularly prevalent in the posterior fossa, and NBL is the most common extracranial solid tumor in children [66,67]. Despite much success, more targeted and effective therapies are needed for children whose tumors cannot be completely cured.

The commonality among different neural tumors can be observed only in the comparison between GBM and MBL (Table S3.23 in S1 Data). It is also worth noting that GBM and MBL cohorts have specific gene sets relative to other three brain tumors (Fig 4D and 4E). For

example, when compared to NBL cohorts (NBL_TARGET and NBL_BROAD), GBM_DE has specific genes, such as *PDGFRB* and *PTPRD* in the PAK pathway ($K = 4$), as well as *ID3* and *CEBPA* ($K = 8$) in the NF-$\kappa$B pathway (Table S3.25 in S1 Data), while MBL_DE_K and MBL_CA feature *ID3*, *GNA11*, and *PRF1* ($K = 10$) in circadian rhythm regulation (Table S3.28 in S1 Data). This suggested *ID3* could serve as a common regulatory gene for GBM and MBL. Unexpectedly, their common genes *PTPRD*, *PDGFRB*, *RBM38*, and *KLHL36* are also identified as overlapping specific genes distinguishing GBM and MBL from NBL. These results may provide insights into potential common targeted therapies.

Despite the similarities between GBM and MBL, we also focused on their molecular differences and successfully identified their specific therapeutic pathways using ModSDP. *CASP8*, *LRP1B*, *PDGFRB*, *RAC1* and *SMO* are representative specific genes for GBM_DE (Table S3.24 in S1 Data) and are involved in the ERK signaling. This pathway can inhibit the degradation of programmed death-ligand 1 (PD-L1) and maintain PD-L1 stability in glioblastoma [68]. On the other hand, we detected a specific gene set for MBL_CA: {*NUMA1*, *CLIP1*, *PTCH1*, *STK11*, *CLTC*}. Among these, *NUMA1* and *CLIP1* act in the cell cycle pathway, while *CLIP1*, *PTCH1*, *STK11*, and *CLTC* affect signaling by GPCR. This suggests that signaling pathway crosstalk may contribute to the development of MBL.

Subtype similarity is also embodied in NBL and PA, since the results of GBM versus PA are the same as those versus NBL when $K$ ranges from 2 to 4 (Tables S3.25 and S3.26 in S1 Data). Compared with two NBL cohorts (NBL_TARGET and NBL_BROAD) and PA_DE, the specific genes of GBM_DE are {*CEBPA*, *SMO*} and {*FOXA1*, *RGL3*}, respectively. However, NBL exhibited relatively more heterogeneity than PA, because NBL (NBL_BROAD) has specific gene sets to PA (PA_DE), while PA lacks specific gene sets relative to any other subtype (Fig 4D). This specificity is observed in *IRS4*, *ATM*, *FBN2*, and *FLT4* (Fig 4E and Table S3.27 in S1 Data, $K = 9$), which are involved in the ERK signaling pathway that has been verified to impact NBL progression [69]. The two MBL cohorts (MBL_DE_K and MBL_CA) also have specific genes compared to PA_DE (Table S3.29 in S1 Data). When $K = 5$, we detected {*FOXA1*, *KLHL36*, *PPP2R1A*, *PTPRD*, *RABEP1*}, among which *PPP2R1A* and *RABEP1* are involved in the regulation of CFTR activity pathway.

Overall, by comparing to PA, we identified subtype-specific pathways in neural tumors: PAK signaling (*PTPRD* and *PDGFRB*) in GBM, regulation of CFTR activity pathway (*PPP2R1A*, *RABEP1*) in MBL, and ERK signaling (*IRS4* and *ATM*) in NBL. In addition, based on the specificity relative to NBL, we predicted *RBM38*, *KLHL36*, and *ID3* as potential subtype-specific genes for GBM and MBL, while *LRP1B* and *PTPN11* appear to be specific to NBL. These predicted genes are promising for follow-up studies on their roles in signaling pathways.

Treatment for pediatric tumors should differ from that for adults, especially in light of mutations specific to children. The specificity among different tumor types also suggests the extent of tumor heterogeneity and the importance of prioritizing for targeted therapy. This underscores the role of personalized therapy, which can be achieved by careful stratification of patient populations.

## EntCDP and ModSDP indicate that environmental factors may promote pathogenicity of cancers through pathways

Smoking, alcohol abuse and high BMI are three tumor-promoting agents that may induce genetic mutations and promote the development of cancers [70,71]. By comparing signaling pathways between smokers and nonsmokers, drinkers and nondrinkers as well as obese and normal-weight individuals with the same type of cancer, we analyzed the relationship

between the three factors and signaling pathways to inform targeted treatments for each group.

**Risk factors differ between smokers and nonsmokers.** It is well established that lung cancer is closely associated with tobacco use [70,72]. The fact that smokers outnumber nonsmokers in cohort LUAD further demonstrates the higher risk of lung cancer among smokers (Fig 5A).

To begin with, we analyzed the common driver pathways shared by all LUAD patients, regardless of their smoking history. Using EntCDP with $K = 9$, we detected {*KRAS*, *RAF1*, *TP53*, *MET*, *CTNNB1*, *PTPN6*, *FOXA1*, *MAX*, *NT5C3A*}. The first three genes are enriched in pathways related to non-small cell lung cancer, while the first four genes are enriched in the central carbon metabolism in cancer, a frequently altered pathway in lung adenocarcinoma [73]. These results demonstrate the effectiveness of EntCDP in capturing functionally meaningful genetic patterns.

However, what causes lung cancer in those patients who have never smoked remains unclear. Notably, when focusing on non-smoking LUAD patients, we identified signaling by GPCR when $K = 9$ (Fig 5 and Table S4.1 in S1 Data). There is reason to believe that *EGFR* plays a prominent role in inducing LUAD through this pathway, since it exhibits a high frequency of mutations and has been identified as a prognostic factor for lung cancer in never smokers (LCINS) [74]. In addition, we presume that the mutated *CYSLTR2* may be involved in lung cancer development through the production of aberrant GPCRs, based on the evidence observed in other cancers [75].

Human life is seriously affected by another smoking-associated cancer: head and neck squamous cell carcinoma (HNSC) [76], the sixth most common cancer worldwide [77]. There are significant differences in signaling pathways between HNSC patients exposed to tobacco and those who are not. In patients without a history of smoking, *RASA1*, *BRAF* and *CACNA1D* may promote the proliferation of head and neck cells by regulating the MAPK signaling pathway (S3A Fig and Table S4.2 in S1 Data). For smokers, we identified *COL1A1*, *PIK3CA*, and *LPAR4* as specific genes in the PI3K-Akt signaling pathway, which has been confirmed to be associated with HNSC [78,79]. We predicted the T-cell leukemia virus-1 (HTLV-1) infection pathway as a novel finding in HNSC because we identified *XPO1* and *EP300* as smoker-specific biomarkers involved in this pathway.

**Shedding light on alcohol exposure.** Beyond tobacco use, alcohol consumption is another major risk factor for head and neck cancer [80]. We found that drinkers generally carry specific genes such as *NOTCH1*, *CDKN2A* and *BRCA1* (S3C Fig and Table S4.4 in S1 Data). Their important role in the GPCR pathway is consistent with evidence that changes in GPCR function are implicated in ethanol exposure [81]. For nondrinkers, we detected {*CASP8*, *FH*, *MAPK1*, *RUNX1*, *STAT3*, *ETV5*, *EXT2*, *NRP1*, *ZNF626*} when $K = 9$. The first five genes are members of the pathways in cancer, among which *CASP8*, *STAT3* and *MAPK1* are related to viral carcinogenesis, including Kaposi's sarcoma-associated herpesvirus and human cytomegalovirus (HCMV) infection. This highlights the need to explore virus-related mechanisms in non-alcohol-associated tumor development.

In addition to HNSC, drinkers make up almost three-quarters of the patients in the cohort ESCA (Fig 5B). In particular, *NOTCH1*, characterized as a drinker-specific gene in HNSC, is also identified in ESCA drinkers, which underlines its core status in alcohol-related cancers [71]. *TP53* also attracts our attention because its high mutation rate exceeds 80% in both alcoholics and nonalcoholics. In addition, the remaining patients who never drink may be affected by a pathway called MHC class I antigen presentation (Fig 5B). The pathogenesis of

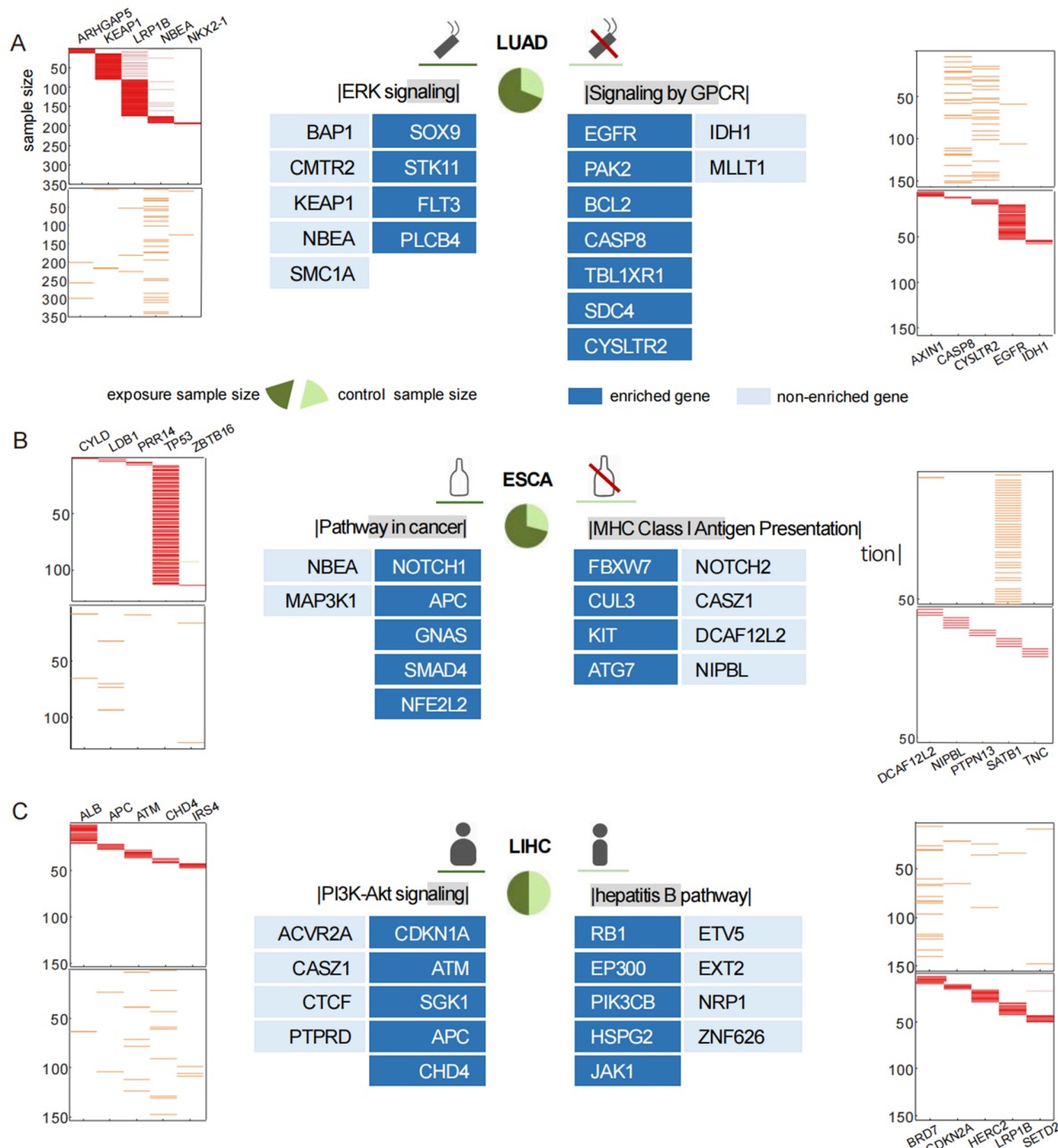

**Fig 5. Three environmental agents promote tumor progression. Specific signaling pathways in patients with risk exposures to tobacco (A), alcohol (B) and high BMI (C) are illustrated on the left side, while the control groups without harmful habits or abnormal indices are shown on the right.** The pie chart in the middle shows the proportions of the two groups. The heatmaps on both sides exhibit the gene mutation profiles of specific genes in the exposure group (the two maps on the left) and the control group (the two maps on the right) when parameter K is 5. Heatmaps in the same horizontal line share the same gene set. Genes enriched in identified pathways in the middle are filled in dark blue.

eosinophilic esophagitis has been reported to depend on presenting antigens on the major histocompatibility complex (MHC) class II [82], which may imply the threat of MHC Class I antigen presentation in esophageal cancers.

**Obesity as an underlying factor disrupting signaling pathways.** It is believed that being overweight may also increase the risk of many cancers, such as stomach adenocarcinoma, liver hepatocellular carcinoma, breast carcinoma, and colorectal cancer [83]. Therefore, we investigated whether signaling pathways can serve as a bridge from obesity to cancer.

It has been reported that the PI3K/Akt/mTOR cascade is a target of many obesity-associated factors that regulate cell proliferation and survival [84]. We verified this result when we focused on patients with hepatocellular carcinoma. We identified *CDKN1A*, *ATM*, *SGK1*, *APC* and *CHD4* (Table S4.7 in S1 Data, *K* = 9) as specific genes of overweight individuals. Enrichment analysis further revealed that these genes can be mapped to the PI3K-Akt signaling pathway. For those of normal weight, the hepatitis B pathway, composed of *RB1*, *EP300*, *PIK3CB*, *HSPG2* and *JAK1* (*K* = 9), may serve as a key point for effective treatment and prevention (Fig 5C), because chronic infection with hepatitis viruses is a major causative factor for hepatocellular carcinoma [85].

Additionally, it is particularly common that people who are obese also tend to have colorectal cancer (S3E Fig). We conclude that mutation in *CDKN1B*, *CDKN2C*, *TP53*, *ATR*, and *CCR7* may increase the risk of developing colorectal cancer through the ERK signaling pathway in overweight individuals. However, those with a normal BMI should be cautious to prevent Yersinia infection [86], since *MAP2K1*, *PLCG1*, and *MYD88* exhibit mutual exclusivity and are significantly enriched in this pathway.

In summary, viral and bacterial infections as causes of human cancer have gained new significance because we identified the HCMV infection, HIV life cycle, hepatitis B, and Yersinia pathway in several control groups. Moreover, we have reason to believe that the Notch-mediated pathway may be responsible for bridging alcohol and cancers, while obesity would increase the risk of cancer through signals regulated by the CDKN gene family. For more details about risk factors for other cancers, please refer to Sect D in S1 Text.

## Discussion

All life on Earth, like the entire ecosystem in which we live, needs to maintain self-homeostasis through intricately complex molecular interactions. As is usually the case, forest fires are not attributed directly to global warming, even when a certain connection is apparent; instead, they are typically ascribed to specific triggers such as human activity and extreme weather events. Similar to the study of natural disasters, scientists are also thinking about the mechanisms of disease progression from local but critical events rather than the entire system. In this context, efforts are being made to compile a comprehensive list of disease-associated biomolecules, such as genes, proteins, mRNAs and related signaling pathways.

Currently, more attention is being paid to cancer, a severely complex disease and the leading cause of death worldwide. Since molecular targeted therapies represent a breakthrough in the management of this devastating illness, verification and functional annotation of driver genes are precious sources for treatment and are continually being supplemented and improved. We are involved in this major project and shift our perspective to the relationship between the signal molecular interrelation, namely, signaling pathway, and the causes of cancer. Therefore, we collect large-scale data on gene mutations in pan-cancer genomes from public databases and apply our two models, EntCDP and ModSDP, to identify common and specific pathways. Unlike traditional approaches that focus on a single cancer type, our comparative analysis enables the discovery of pathways shared by, or specific to, more than one type of cancer.

Our method addresses the shortcomings of the original models and improves the accuracy of signaling pathways identification. EntCDP solves the problem of inaccurate searches

caused by the large variations in mutation rates across different datasets, while ModSDP adjusts the coverage and exclusivity of signaling pathways to achieve a balanced consideration of both aspects [14]. At the same time, the signaling pathways we identify are constituted by more mutually exclusive alterations, which exert stronger selective pressure on tumors.

Among the tens of thousands of genes that a human cell carries, our identification scope is deliberately narrowed down to 568 reliable reference genes derived from an integrated model (IntOGen). Therefore, our results not only boost the efficiency of model execution, but also prevent genes not related to cancer from being falsely assumed to regulate oncogenic signals. Additionally, several cohort-sensitive genes identified by IntOGen are also recovered by our models, which supports the reliability of our findings.

In clinical practice, molecular subtypes can guide prognosis and therapeutic treatment in many cancer types. Beyond histological subtyping, we expect that individual patient information and clinical indicators can also be used as criteria for clinical decision-making, which is also the original intention of our study. Naturally, our two models have the advantage of categorical comparison.

On the one hand, common and specific signals among different types or subtypes of cancer can be easily identified for systematic analysis and interpretation. For example, EntCDP identified *TP53* as an important gene implicated in gastrointestinal disorders. When looking at women's cancers, disruption of the ErbB signaling pathway may contribute to breast and ovarian cancer. ModSDP identified the mTOR signaling pathway in lung squamous cell carcinoma and the FoxO signaling in lung adenocarcinoma, which can help with understanding subtype heterogeneity. On the other hand, we use the information provided by the data to subdivide samples of the same cancer according to the region, age, and risk factors. Region-specific analysis reveals that PI3K-Akt signaling pathway regulated by *HLA-A* and *HLA-B* is involved in Chinese patients with bladder cancer compared to Americans. In contrast, we predicted that therapeutic strategies for American patients might prioritize genes in signaling by GPCR, including *CDKN1A*, *IRS4*, and *KMT2D*. One ingenuity of our investigation in pediatric tumors is that we not only compare different cancers within the pediatric group, but also contrast children and adults with the same cancers (such as GBM and AML). Specifically, we found that the Ras signaling pathway, represented by *NRAS*, *KIT*, *PTPN11*, and *KRAS*, plays an important role in children with AML. Similarly, *CEBPA* and *ID3* are identified through NF-$\kappa$B signaling as potential drivers in pediatric GBM. Risk factor analysis underscores viral and bacterial infections (HIV, HBV, Yersinia) in cancer etiology. Additionally, we hypothesize that Notch signaling mediates alcohol-related carcinogenesis, while obesity may promote cancer via CDKN-regulated pathways.

We have considered further subdividing the data to provide a more precise reference for targeted therapy, but the application of our models is limited by the size of the data. In other words, our models can only search for more accurate signal systems with larger sample sizes. Specifically, limited sample sizes and lack of subtype data in pediatric patients are challenges to constructing significant and sophisticated mutation profiles for young patients, making it impossible to further account for the stage of tumor and subdivisions of childhood age. In addition, other divisions of datasets can also be explored. For example, it is also worth considering the races and continents not included in our study to better address the complexity and scope of the issues involved.

Our study highlights the need for classification and refinement of cancer patients. In addition to the reference lists provided in this paper, our models are applicable to other comparable populations. Finding the appropriate reference pathways and targets for clinical patients who best match their population marks a step forward in cancer genomics from bench to bedside.

## Materials and methods

### Data collection and processing

We obtained pan-cancer genomes across 23 tumor types from TCGA (https://portal.gdc.cancer.gov/), ICGC (https://dcc.icgc.org/releases/current/Projects/), cBioPortal (https://www.cbioportal.org/) and PCAWG (https://xenabrowser.net/datapages/?hub=https://pcawg.xenahubs.net:443). A total of 16,103 sequencing samples, consisting of 13,535 adult and 1,539 pediatric tumors, were collated into 55 datasets for analysis. We also collected clinical data for these samples to investigate clinicopathological characteristics, including region, age, smoking history, drinking history, and obesity.

Most TCGA cancer patients are adults, so we only selected patients aged 18 years or older to compare with children, while retaining their original cohort name. Similarly, for the pediatric cohorts, we eliminated samples older than 18, ensuring all pediatric patients included in our study are under 18 years old. According to the WHO criteria, individuals with a BMI greater than 25 were considered obese in our study.

### EntCDP: Entropy-based common driver pathway identification model

EntCDP takes as input $R$ gene mutation matrices $A_r = (a_{ij}^r)_{m_r \times G_r} (r = 1, 2, \ldots, R \geq 2)$, where $m_r$ is the number of samples and $G_r$ is the number of genes in $A_r$. $a_{ij}^r$ indicates the mutation status of the $j$-th gene in the $i$-th sample in the $r$-th matrix, with a value of 1 (mutation) or 0 (no mutation). First, the number of columns is expanded to $|G| = | \cup_{r=1}^R G_r'|$, where $G_r'$ denotes the set of genes in $A_r$ and $| \cdot |$ represents the cardinality, so that the same column in each matrix corresponds to the same gene. If a gene corresponding to a column is not present in a particular matrix, all entries in that column of the matrix are set to zero. To identify a common mutated driver gene set $M$ containing $K$ genes with large coverage and high exclusivity among the $R$ cancers (i.e., $M$ exhibits large coverage and high exclusivity in each cancer dataset $A_r$), EntCDP introduces an objective function $C_m(M)$:

$$C_m(M) = \omega \cdot \sum_{r=1}^R \left[ 2|\Gamma_{A_r}(M)| - \sum_{g \in M} |\Gamma_{A_r}(g)| \right], \tag{1}$$

where $\Gamma_{A_r}(g)$ denotes the set of samples in $A_r$ with gene $g$ mutated, and $|\Gamma_{A_r}(M)| = |\cup_{g \in M} \Gamma_{A_r}(g)|$ represents the coverage of $M$ in $A_r$. In addition, $\omega = H(\{CR_r^{(M)}, r = 1, \ldots, R\}) = -\sum_{r=1}^R (CR_r^{(M)} / \sum_{r=1}^R CR_r^{(M)}) \log(CR_r^{(M)} / \sum_{r=1}^R CR_r^{(M)})$, where the coverage rate $CR_r^{(M)} = |\Gamma_{A_r}(M)|/m_r$ and $m_r$ is the sample size of $A_r$. As $\omega$ increases, the variation in the coverage of $M$ across different cancer types decreases. We use genetic algorithms to maximize Eq 1. For more details about EntCDP, please refer to [14].

### ModSDP: Modified specific driver pathway identification model

ModSDP takes as input two groups of gene mutation matrices: $A_r = (a_{ij}^r)_{m_r \times G_r} (r = 1, 2, \ldots, R)$ and $B_t = (b_{ij}^t)_{m_t \times G_t} (t = 1, 2, \ldots, T)$, where $m_r$ and $G_r$ are the number of samples and genes in $A_r$, and $m_t$ and $G_t$ are the number of samples and genes in $B_t$. Similarly, the number of columns of all matrices is expanded to the same, so that each column corresponds to the same gene across all matrices. To identify a specific mutated driver gene set $M$ for the $R$ cancer types relative to the $T$ types (i.e., $M$ exhibits large coverage and high exclusivity in each $A_r$ but lacks these properties in each $B_t$), ModSDP introduces an objective function $S_m(M)$:

$$S_m(M) = \frac{1}{R} \sum_{r=1}^{R} \left[ 2|\Gamma_{A_r}(M)| - \sum_{g \in M} |\Gamma_{A_r}(g)| \right]$$
$$- \frac{1}{T} \sum_{t=1}^{T} \left[ K|\Gamma_{B_t}(M)| - \sum_{g \in M} |\Gamma_{B_t}(g)| \right] / (K - 1), \tag{2}$$

where $K$ is the number of genes in $M$. We maximize the binary form of Eq 2 using IBM ILOG CPLEX optimizer, a high-performance mathematical programming solver. Further details about ModSDP are provided in [14].

## Permutation test

A gene set identified by EntCDP and ModSDP should be subjected to a permutation test to check whether it truly exhibits high coverage and high exclusivity characteristics of a signaling pathway. We compute two types of significance: individual significance, which evaluates a gene set within a single mutation matrix using statistics such as $2|\Gamma_{A_r}(M)| - \sum_{g \in M}|\Gamma_{A_r}(g)|$; and overall significance, which assesses a gene set across all matrices using the weights $C_m(M)$ in Eq 1 for EntCDP and $S_m(M)$ in Eq 2 for ModSDP.

The permutation test follows the approach in [12], where $p$-values are calculated based on 1,000 random permutations of the original mutation matrices. Each permutation preserves the mutation frequency of genes but randomized their sample assignments, thereby disrupting original mutation patterns. For EntCDP, significance is confirmed when both individual and overall $p$-values satisfy $p<0.05$. For ModSDP, overall significance also requires $p<0.05$, while individual significance requires $p<0.05$ for each $A_r$ and $p \geq 0.05$ for each $B_t$.

The results of each comparison are presented in individual tables in S1 Data, listing significant driver gene sets containing 2 to 10 genes.

## Pathway enrichment and gene set prioritization

We performed pathway enrichment analysis for each significant gene set by DAVID tool [87] to evaluate their potential enrichment in the Kyoto Encyclopedia of Genes and Genomes (KEGG) and Gene Ontology (GO) databases. We set 0.05 as the significance threshold.

Subsequently, for each gene set enriched in significant pathways, we quantified the enrichment ratio as the maximum proportion of genes mapped to any enriched pathway. The gene set with the highest enrichment ratio was selected as the final candidate for downstream analysis.

## Supporting information

**S1 Text. Supporting information file including additional methods, analyses on regional bias, similar cancers, and risk factors.**
(PDF)

**S1 Fig. Venn diagrams highlight the overlap (dark region) in hypergeometric test for commonality between BRCA and BRCA_UK (A) and specificity of BRCA relative to BRCA_UK (B).**
(TIF)

**S2 Fig. Comparison of COADREAD and LIHC. (A) The result of common/specific gene sets of COADREAD and LIHC by EntCDP/ModSDP. (B) Regulation of genes involved in the signaling pathways shown in (A).**
(TIF)

**S3 Fig. Environmental agents promote tumor progression. Specific signaling pathways in patients with risk exposures to tobacco (A, B), alcohol (C, D) and high BMI (E) are illustrated on the left side, while control groups without harmful habits or abnormal indices are on the right.**
(TIF)

**S1 Table. Sample characteristics of cohorts.**
(XLSX)

**S2 Table. Driver genes identified by EntCDP and the comparison with those from IntOGen.**
(XLSX)

**S3 Table. Driver genes identified by ModSDP and the comparison with those from IntOGen.**
(XLSX)

**S1 Data. All tables of the results.**
(PDF)

## Author contributions

**Conceptualization:** Junhua Zhang.

**Data curation:** Wenjia Zhou.

**Formal analysis:** Wenjia Zhou, Junhua Zhang.

**Funding acquisition:** Junhua Zhang.

**Investigation:** Wenjia Zhou.

**Methodology:** Wenjia Zhou, Junhua Zhang.

**Project administration:** Junhua Zhang.

**Resources:** Wenjia Zhou, Junhua Zhang.

**Software:** Wenjia Zhou, Junhua Zhang.

**Supervision:** Junhua Zhang.

**Visualization:** Wenjia Zhou.

**Writing – original draft:** Wenjia Zhou, Junhua Zhang.

**Writing – review & editing:** Wenjia Zhou, Junhua Zhang.

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
