## [Decision Letter · Decision Letter 0]

6 Apr 2025

PCOMPBIOL-D-24-02048

Identification of driver pathways to explore commonalities and specificities across 23 cancer types: a comprehensive study incorporating region, age, and risk factors

PLOS Computational Biology

Dear Dr. Zhang,

Thank you for submitting your manuscript to PLOS Computational Biology. We apologize for the delay in processing your submission. After careful consideration, we feel that it has merit but does not fully meet PLOS Computational Biology's publication criteria as it currently stands. Therefore, we invite you to submit a revised version of the manuscript that addresses the points raised during the review process.

Please submit your revised manuscript within 60 days Jun 06 2025 11:59PM. If you will need more time than this to complete your revisions, please reply to this message or contact the journal office at ploscompbiol@plos.org. Please include the following items when submitting your revised manuscript:

We look forward to receiving your revised manuscript.

Kind regards,

Melissa L. Kemp, Ph.D.

Academic Editor

PLOS Computational Biology

Ilya Ioshikhes

Section Editor

PLOS Computational Biology

**Journal Requirements:**

1) Please ensure that the CRediT author contributions listed for every co-author are completed accurately and in full.At this stage, the following Authors/Authors require contributions: Wenjia Zhou, and Junhua Zhang. Please ensure that the full contributions of each author are acknowledged in the "Add/Edit/Remove Authors" section of our submission form.The list of CRediT author contributions may be found here: https://journals.plos.org/ploscompbiol/s/authorship#loc-author-contributions 2) We ask that a manuscript source file is provided at Revision. Please upload your manuscript file as a .doc, .docx, .rtf or .tex. If you are providing a .tex file, please upload it under the item type u2018LaTeX Source Fileu2019 and leave your .pdf version as the item type u2018Manuscriptu2019. 3) Please insert an Ethics Statement at the beginning of your Methods section, under a subheading 'Ethics Statement'. It must include:i) The full name(s) of the Institutional Review Board(s) or Ethics Committee(s)ii) The approval number(s), or a statement that approval was granted by the named board(s)iii) A statement that formal consent was obtained (must state whether verbal/written) OR the reason consent was not obtained (e.g. anonymity). NOTE: If child participants, the statement must declare that formal consent was obtained from the parent/guardian.]. 4) Please upload all main figures as separate Figure files in .tif or .eps format. For more information about how to convert and format your figure files please see our guidelines: https://journals.plos.org/ploscompbiol/s/figures 5) Please upload a copy of Subfigures 3G, and 3H which you refer to in your text on page 10. Or, if they are no longer to be included as part of the submission please remove all reference to them within the text. 6) We have noticed that you have uploaded Supporting Information files, but you have not included a complete list of legends. Please add a full list of legends for your Supporting Information file (Source code for EntCDP and ModSDP.rar) after the references list. 7) Some material included in your submission may be copyrighted. According to PLOSu2019s copyright policy, authors who use figures or other material (e.g., graphics, clipart, maps) from another author or copyright holder must demonstrate or obtain permission to publish this material under the Creative Commons Attribution 4.0 International (CC BY 4.0) License used by PLOS journals. Please closely review the details of PLOSu2019s copyright requirements here: PLOS Licenses and Copyright. If you need to request permissions from a copyright holder, you may use PLOS's Copyright Content Permission form.Please respond directly to this email and provide any known details concerning your material's license terms and permissions required for reuse, even if you have not yet obtained copyright permissions or are unsure of your material's copyright compatibility. Once you have responded and addressed all other outstanding technical requirements, you may resubmit your manuscript within Editorial Manager. Potential Copyright Issues:i) Figures 4, 5, and C. Please confirm whether you drew the images / clip-art within the figure panels by hand. If you did not draw the images, please provide (a) a link to the source of the images or icons and their license / terms of use; or (b) written permission from the copyright holder to publish the images or icons under our CC BY 4.0 license. Alternatively, you may replace the images with open source alternatives. See these open source resources you may use to replace images / clip-art:- https://commons.wikimedia.org- https://openclipart.org/.  8) Please amend your detailed Financial Disclosure statement. This is published with the article. It must therefore be completed in full sentences and contain the exact wording you wish to be published.1) State the initials, alongside each funding source, of each author to receive each grant. For example: "This work was supported by the National Institutes of Health (####### to AM; ###### to CJ) and the National Science Foundation (###### to AM)."2) State what role the funders took in the study. If the funders had no role in your study, please state: "The funders had no role in study design, data collection and analysis, decision to publish, or preparation of the manuscript."3) If any authors received a salary from any of your funders, please state which authors and which funders.. 

**Reviewers' comments:**

Reviewer's Responses to Questions

Reviewer #1: In this manuscript, the authors conduct a comprehensive analysis of multiple cancer omics data by focusing on the identificaiton of cancer driver genes and pathways. Overall, the manuscript is informative. However, the current manuscript should be revised before acceptance for publication.

1. The current tiltle and abstract are too long, preventing readers from comprehending the key points. The authors should rewrite the title and abstract to make sure that the most novel findings are clearly presented.

2. Findings related to regions, ages, and environmental risky factors are comprehensive, but hard to be validated experimentally. The authors should prioritize the most important and novel points from these comprehensive findings. It is best to form a testable hypothesis.

3. The current figures are hard to follow by readers without reading the legends. The authors should replot the figures and add more labels to improve the readability.

Reviewer #2: Zhou et al. apply their recently developed two models, EntCDP and ModSDP to analyze mutation data. The author provides a comprehensive framework for understanding cancer signaling heterogeneity across diverse populations and clinical contexts. This is a compelling area of research, and I recommend publication with the following revisions.

1. What’s the differences between Ar and Bt in ModSDP model? How to determine the number of driver genes K? Further elaboration on these points would enhance clarity and comprehensibility.

2. Could the method be applicable for identifying driver pathways across different subtypes of one cancer? For example, Luminal A, Luminal B, Basal of breast cancer.

**Have the authors made all data and (if applicable) computational code underlying the findings in their manuscript fully available?**

Reviewer #1: None

Reviewer #2: None

PLOS authors have the option to publish the peer review history of their article (what does this mean?). If published, this will include your full peer review and any attached files.

Reviewer #1: No

Reviewer #2: No

**Figure resubmission:**
---

## [Decision Letter · Decision Letter 1]

15 Jun 2025

PCOMPBIOL-D-24-02048R1

Multi-context modeling of driver pathways reveals common and specific mechanisms across 23 cancer types

PLOS Computational Biology

Dear Dr. Zhang,

Thank you for submitting your manuscript to PLOS Computational Biology. After careful consideration, we feel that it has merit but does not fully meet PLOS Computational Biology's publication criteria as it currently stands. Therefore, we invite you to submit a revised version of the manuscript that addresses the points raised during the review process.

Please note that at this stage, the reviewers have responded that you have addressed all of their scientific concerns. However I note as an editor that this manuscript still suffers from a number of grammatical and language issues that I would like to see resolved before acceptance of a final form of the paper. Please work with an English language assistance at your institution to reread and make final edits before we move to full acceptance of the paper and type-setting stage.

Please submit your revised manuscript within 30 days Aug 15 2025 11:59PM. If you will need more time than this to complete your revisions, please reply to this message or contact the journal office at ploscompbiol@plos.org. Please include the following items when submitting your revised manuscript:

We look forward to receiving your revised manuscript.

Kind regards,

Melissa L. Kemp, Ph.D.

Academic Editor

PLOS Computational Biology

Ilya Ioshikhes

Section Editor

PLOS Computational Biology

**Journal Requirements:**

1) We have noticed that you have a list of Supporting Information legends in your manuscript for Supplementary Figures (1-3). However, there are no corresponding files uploaded to the submission. Please upload them as separate files with the item type 'Supporting Information'.

2) We noted that the main figures are included in the "Files for compiling Revised_Manuscript.zip"  with the Item Family "Supplemental." Please note that the main figures should be uploaded as separate figure files in .tif or .eps format with the Item Family "Figure."

**Reviewers' comments:**

Reviewer's Responses to Questions

Reviewer #1: The authors have revised the manuscript according to the reviewers' comments and the manuscript is acceptable now.

Reviewer #2: The revisions made are satisfactory.

**Have the authors made all data and (if applicable) computational code underlying the findings in their manuscript fully available?**

Reviewer #1: Yes

Reviewer #2: None

PLOS authors have the option to publish the peer review history of their article (what does this mean?). If published, this will include your full peer review and any attached files.

Reviewer #1: No

Reviewer #2: No

**Figure resubmission:**
---

## [Editor Report · Decision Letter 2]

21 Jul 2025

Dear Dr Zhang,

We are pleased to inform you that your manuscript 'Multi-context modeling of driver pathways reveals common and specific mechanisms across 23 cancer types' has been provisionally accepted for publication in PLOS Computational Biology.

Best regards,

Melissa L. Kemp, Ph.D.

Academic Editor

PLOS Computational Biology

Ilya Ioshikhes

Section Editor

PLOS Computational Biology

---

## [Editor Report · Acceptance letter]

PCOMPBIOL-D-24-02048R2

Multi-context modeling of driver pathways reveals common and specific mechanisms across 23 cancer types

Dear Dr Zhang,

I am pleased to inform you that your manuscript has been formally accepted for publication in PLOS Computational Biology. Your manuscript is now with our production department and you will be notified of the publication date in due course.

With kind regards,

Zsofia Freund
